# Dietary Supplementation with Putrescine Improves Growth Performance and Meat Quality of Wenchang Chickens

**DOI:** 10.3390/ani13091564

**Published:** 2023-05-07

**Authors:** Qi Qi, Chengjun Hu, Haojie Zhang, Ruiping Sun, Quanwei Liu, Kun Ouyang, Yali Xie, Xiang Li, Wei Wu, Yuhang Liu, Guiping Zhao, Limin Wei

**Affiliations:** 1Sanya Institute, Hainan Academy of Agricultural Sciences (Hainan Experimental Animal Research Center), Sanya 572025, China; 2Hainan Key Laboratory of Tropical Animal Breeding and Epidemic Research, Institute of Animal Husbandry & Veterinary Research, Hainan Academy of Agricultural Sciences, Haikou 571100, China; 3Tropical Crop Genetic Resource Research Institute, Chinese Academy of Tropical Agricultural Sciences, Haikou 571101, China; 4College of Ocean and Fishery, Guangdong Eco-Engineering Polytechnic, Guangzhou 510220, China

**Keywords:** amino acid, fatty acid, growth performance, putrescine, Wenchang chickens

## Abstract

**Simple Summary:**

Putrescine plays a vital role in various biological processes. However, there is little emphasis on the effect of putrescine on meat quality in broilers. This study demonstrated that broilers fed diet supplemented with 0.05% putrescine showed higher average daily gain. In addition, dietary supplementation with 0.05% putrescine increased the contents of essential amino acids but reduced the saturated fatty acids level in muscle. These results indicated that dietary supplementation with putrescine is an effective strategy to improve the growth performance and meat quality of broilers.

**Abstract:**

This study was to investigate the effects of dietary supplementation with putrescine on the growth performance and meat quality of chickens. A total of 480 eighty-day-old female Wenchang chickens were randomly assigned into four groups, with 8 replications per group and 15 animals per replicate. The chickens in the control group (Con) were fed a basal diet; the 3 experimental groups were fed a basal diet with 0.01%, 0.03%, and 0.05% putrescine, respectively. The experiment lasted for 40 days. The results showed that dietary supplementation with 0.05% putrescine increased (*p* < 0.05) the final body weight and average daily weight gain, and decreased the ratio of feed intake to the body weight gain of Wenchang chickens. Dietary supplementation with putrescine decreased the concentrations of putrescine, spermidine, and spermine in serum (*p* < 0.05). The contents of methionine, phenylalanine, lysine, aspartic acid, tyrosine, total essential amino acids, and total amino acids in breast muscle were higher (*p* < 0.05) in 0.03% and 0.05% groups than those in Con group. However, the contents of undecanoic acid, lauric acid, tridecanoic acid, myristic acid, pentadecanoic acid, arachidic acid, docosanoic acid, tricosanic acid, lignoceric acid, erucic acid, cis-11,14,17-eicosatrienoate, linoleic acid, and total n-6 monounsaturated fatty acids in breast muscle were lower (*p* < 0.05) in 0.03% and 0.05% groups than those in Con group. In addition, putrescine supplementation decreased (*p* < 0.05) the ratio of n-6/n-3 polyunsaturated fatty acids in breast meat. Overall, dietary supplementation with 0.05% putrescine enhanced the growth performance and meat quality of Wenchang chickens.

## 1. Introduction

Chicken meat is not only an ideal protein source for humans, but also contains health-promoting substances, such as essential amino acids and n-3 polyunsaturated fatty acids [1,2]. Broilers are reared at a high stocking density to increase the production efficiency in modern poultry industry. However, high stocking density was shown to decrease the meat quality of broilers [3]. In addition, physiological stress levels were amplified under intensive rearing environments, resulting in poor meat quality traits, such as increased muscle shear force and decreased intramuscular fat [4,5,6]. Therefore, improving meat quality is of great significance to promoting human health and the development of the poultry industry.

Putrescine exerts a vital role in various biological processes, including promoting cell proliferation and protein synthesis, and alleviating oxidative stress [7,8]. In weaning piglets, diets supplemented with putrescine increased the antioxidant enzyme activities in the liver and intestine [9]. In addition, putrescine is the precursor of spermine [10]. A previous study showed that spermine can inhibit the cell damage caused by higher reactive oxygen species production and endoplasmic reticulum stress [11]. Increased oxidative stress is known to negatively affect meat quality [12]. These results indicate that putrescine has the potential to improve meat quality via alleviating oxidative stress. However, a high concentration of putrescine is toxic, leading to a toxicological reaction in cells and birds [13,14].

Based on these results, we hypothesize that dietary supplementation with an appropriate content of putrescine could improve the meat quality of broilers. Therefore, the objective of the present study was to evaluate the effects of dietary supplementation with putrescine on the growth performance and meat quality of broilers.

## 2. Materials and Methods

### 2.1. Experimental Design and Diets

A total of 480 80-day-old female broilers (Wenchang chickens) were randomly divided into 4 experimental groups, with 8 replicates per group and 15 broilers per replicate. The chickens in the control group (Con) were fed a basal diet; the 3 experimental groups were fed a basal diet with 0.01%, 0.03%, and 0.05% putrescine, respectively. The experiment lasted for 40 days. The basal diet (Table 1) was designed in accordance with the Nutrient Requirements of Yellow-feathered Broiler (Ministry of Agriculture, 2004) (NY/T 33-2004). All birds were kept in an environmentally controlled room with artificial light and natural light, and access to feed and water *ad libitum*. All chickens were routinely immunized. The temperature of the room was kept at about 32 °C and the humidity was 55–65%. The feed intake was recorded daily during the experiment, and the initial body weight (IW) and final body weight (FW) were recorded at days 1 and 41 of the experiment. The average daily gain (ADG), average daily feed intake (ADFI), and the ratio of feed intake to body weight gain (F:G) were calculated based on body weight and feed intake.

### 2.2. Sample Collection

At the end of the experiment, 8 chickens close to final BW were selected for blood and muscle collection in each group. About 5 mL of blood samples were taken aseptically from the jugular vein and centrifuged at 4000× *g* for 10 min at 4 °C. Serum was separated and stored at −20 °C for further analysis. After blood sampling, both sides of breast muscle were removed from the carcass according to the standard method of dissection. Mid-segments of left breast muscles were collected and cut into sections fixed in 4% paraformaldehyde solution. The other part of the breast muscle was immediately snap-frozen in liquid nitrogen and then stored at −80 °C for further analysis.

### 2.3. The Level of Polyamine in Serum and Muscle

Muscle or serum samples were homogenized in 4 M perchloric acid solution at −20 °C overnight. The lysate was centrifuged at 8000 r·min^−1^ at 4 °C for 10 min to discard precipitation, and centrifuged again under the same conditions. The supernatant fluid was filtered by a 0.22 µm filtration membrane. The content of polyamine was determined using high performance liquid chromatography methods, as described previously [15].

### 2.4. Meat Quality

The pH of breast muscle was measured at 45 min and 24 h postmortem using a handheld pH meter (FE28-Standard, Mettler-Toledo, Zurich, Switzerland) by inserting the electrode into the muscle. The lightness (L*) and yellowness (b*) were measured using a Chroma Meter (TS7700, Group 3NH, Shenzhen, China) according to the manufacturer’s instructions. About 5 g of breast muscle sample was placed in a polyethylene container and reweighed after being hung at 4 °C for 24 h. The drip loss was calculated as a percentage of the initial weight. Another part of the breast muscle was cut into 2 cm (length) × 1 cm (width) × 0.5 cm (height) parallel to the muscle fibers, and the shear force was measured using a meat shear machine (C-LM36, Northeast Agricultural University, Harbin, China).

### 2.5. Muscle Morphology

The fixed breast muscle was cut into sections with a thickness of 5 µm and stained with hematoxylin-eosin. Five photographs were randomly selected for each slide. The diameter of muscle fiber and total cross-sectional area (μm^2^) of the breast muscle were analyzed using Image-proplus 6.0 software.

### 2.6. Inosine Monophosphate (IMP) Content in Muscle

The muscle samples were cut into slices, dried in a vacuum freeze dryer, and then ground into powder. The content of IMP in muscle was determined using high performance liquid chromatography (LC-1100, Agilent, Santa Clara, CA, USA), as described previously [16]. Briefly, 50 mg of freeze-dried samples were homogenized with 2 mL of phosphate buffer solution (pH = 7.2). A mixed sample and 500 µL of acetonitrile were added into a 2 mL centrifuge tube for an ultrasound for 30 min, and then centrifuged at 5000 r/min for 15 min. The supernatant was taken through a 0.22 µm filtration membrane to a 1.5 mL sample bottle.

### 2.7. Amino Acid and Fatty Acid Contents in Muscle

Approximately 30 mg of freeze-dried samples were hydrolyzed in 6 M hydrochloric acid at 110 °C for 24 h. The suspension was diluted with water, and 1 mL of sample was filtered using a 0.2 µm filter [17]. The samples were analyzed using an HPLC-based automatic amino acid analyzer (Ultimate3000-API 3200 Q TRAP; ThermoFisher Scientific, Waltham, MA, USA), according to the manufacturer’s instructions. The contents of amino acids in the muscle were expressed as mg/g of dried tissue.

Approximately 50 mg of freeze-dried powder samples were mixed with 3 mL of chloroform–methanol–water (volume ratio: 8:4:3). The mixture was shaken for 1 min, followed by an ultrasound in an ice bath for 3 to 5 times, and then centrifuged at 1500 rpm for 15 min. The chloroform layer was transferred to another 10 mL glass centrifuge tube and dried in a vacuum drying oven to obtain fatty acid glyceryl ester. The fatty acid glyceryl ester was analyzed using an automatic fatty acid analyzer (GC-MS 7890B-5977A, Agilent, Santa Clara, CA, USA), as previously described [17].

### 2.8. Elemental Composition in Muscle

The contents of K, Mg, Na, and Zn in breast muscle were analyzed according to the previously described method [18]. Briefly, 0.5 g of freeze-dried sample was digested with 4 mL of nitric acid (65%). Additionally, 1 mL of hydrogen peroxide and 1 mL of hydrofluoric acid were added to the digestion solution, and continuously digested in a constant temperature drying oven at 140~160 °C for 6 h. The filtered sample solution was injected into the ICP-mass spectrometer (ICP-MS, Agilent-7900, Santa Clara, CA, USA), and the concentration of each element was calculated based on the standard curve.

### 2.9. Statistical Analysis

All data were analyzed using SPSS version 26.0 (IBM Corp., Chicago, IL, USA). Data were expressed as means. The differences between groups were analyzed using one-way ANOVA. Duncan’s multiple range test was used for multiple comparisons, with *p* < 0.05 considered as statistically significant.

## 3. Results

### 3.1. Growth Performance

As shown in Table 2, dietary supplementation with 0.05% putrescine increased (*p* < 0.05) the FW and ADG when compared with the Con group. Chickens fed a diet with 0.01% or 0.05% putrescine supplementation converted their feed more efficiently (*p* < 0.05) than those receiving a basal diet. No difference (*p* > 0.05) was observed in the ADFI among the four groups.

### 3.2. The Content of Polyamine in Serum and Muscle

The concentrations of spermine, spermidine, and putrescine in serum in the 0.03% and 0.05% groups were lower (*p* < 0.05) than those in the Con group (Figure 1A–C). Dietary supplementation with 0.01% putrescine decreased (*p* < 0.05) the concentration of putrescine in serum compared to the Con group.

There was no difference (*p* > 0.05) in the contents of spermine, spermidine, and putrescine in breast muscle among all groups (Figure 1D–F).

### 3.3. Carcass Traits

As shown in Table 3, no influence (*p* > 0.05) was observed in the carcass yield, semi-eviscerated percentage, eviscerated percentage, abdominal fat percentage, breast muscle percentage, and thigh muscle percentage among the groups.

### 3.4. Meat Quality and Muscle Morphology

As shown in Table 4, no difference (*p* > 0.05) was found in the pH_45min_, pH_24h_, meat color, drip loss, and shear force of breast muscle among the four groups. Dietary supplementation with putrescine has no effect (*p* > 0.05) on the diameter and area of breast muscle fibers (Figure 2).

### 3.5. Amino Acid Profiles

The contents of methionine (Met), phenylalanine (Phe), lysine (Lys), aspartic acid (Asp), tyrosine (Tyr), total essential amino acids (EAAs), and total amino acids (TAAs) in breast muscle were higher (*p* < 0.05) in the 0.03% and 0.05% putrescine groups than those in the Con group. In addition, dietary supplementation with 0.03% putrescine increased (*p* < 0.05) the contents of threonine (Thr) and proline (Pro) in breast muscle (Table 5).

### 3.6. Fatty Acid Profiles

The contents of undecanoic acid (C11:0), lauric acid (C12:0), tridecanoic acid (C13:0), myristic acid (C14:0), pentadecanoic acid (C15:0), arachidic acid (C20:0), docosanoic acid (C22:0), tricosanic acid (C23:0), lignoceric acid (C24:0), erucic acid (C22:1n9), cis-11,14,17-eicosatrienoate (C20:3n3), linoleic acid (C18:2n6c), and total n-6 monounsaturated fatty acids were lower (*p* < 0.05) in the 0.03% and 0.05% putrescine groups than those in the Con group. Dietary supplementation with putrescine decreased (*p* < 0.05) the n-6/n-3 PUFA ratio in breast muscle. In addition, dietary supplementation with 0.03% putrescine decreased (*p* < 0.05) the content of cis-8,11,14-eicosatrienoate (C20:3n6) in breast muscle (Table 6).

### 3.7. IMP and Elemental Composition

The content of sodium in breast muscle was higher (*p* < 0.05) in the 0.01% and 0.03% groups than that in the Con or 0.05% group. There was no difference (*p* > 0.05) in the contents of IMP, potassium, magnesium, or zinc in breast muscle among all groups (Table 7).

## 4. Discussion

Putrescine plays an important role in cell development and protein synthesis [19]. However, the role of putrescine in the growth performance and meat quality of Wenchang chickens is still unclear. In the present study, a dietary supplementation of 0.05% putrescine increased the FW and ADG by 3.78% and 11.18%, respectively, and decreased the ratio of F:G by 3.65%, whereas had no effect on carcass trait. In line with these results, a previous study reported that dietary supplementation with putrescine increased the final BW and ADG of weaning piglets [20]. Polyamine was shown to promote muscle cell proliferation, growth, and protein synthesis via regulating the gene expression and mTORC1 signaling pathway [21]. In addition, a previous study reported that increased breast muscle mass was accompanied by decreased levels of putrescine and its upstream metabolites [22]. These results suggested that putrescine has the potential to promote muscle growth and development. To further investigate whether dietary supplementation with putrescine could improve the muscle growth and development of Wenchang chickens, the muscle weight and fiber size were determined. Our study showed that no difference was observed in the muscle weight or muscle fiber size of Wenchang chickens among the four groups. This experiment only lasted for 40 days. We suspected that the long-term intake rather than the short-term intake of putrescine can promote muscle development. However, further investigations are needed.

Polyamine plays a crucial role in maintaining the biosynthesis of itself. To investigate the role of putrescine treatment in polyamine metabolism, the concentrations of spermine, spermidine, and putrescine in serum and muscle were determined. In the present study, putrescine treatment increased polyamine metabolism. In contrast, a previous study observed that the long-term intake of polyamine-rich foods gradually increases blood polyamine levels in mice, whereas the short-term intake does not have such effect [23]. This discrepancy between the previous study and our results might be explained by the differences in the animal model used. Interestingly, polyamine content in breast muscle was not affected by dietary putrescine treatment, which suggested that polyamine homeostasis in muscle is substantially independent of dietary supplement. Similar to the present results, Mogridge et al. demonstrated that the contents of putrescine, spermidine, and spermine in the small intestinal of chicks were not affected by dietary putrescine supplementation [24]. A recent study has shown that the major source of polyamine in the blood and muscle is likely derived from the kidney [25]. However, whether polyamine metabolism in the kidney was regulated by putrescine supplementation is still unknown, and further investigation is needed.

Chicken meat provides amino acid for humans. Therefore, the amino acid content in muscle was determined. We found that 0.03% and 0.05% putrescine supplementation had a beneficial effect in improving the protein quality and nutrition value of meat. Gly, Ala, Asp, Glu, Phe, and Tyr are referred to collectively as the flavor amino acid, which can greatly improve the taste of meat [26]. The dietary addition of 0.03% and 0.05% putrescine increased the contents of Tyr, Phe, and Asp in breast meat, which indicates that putrescine supplementation could improve the flavor of chicken meat by increasing the contents of flavor amino acids. Evidence showed that polyamine could increase the mRNA expression levels of amino acid transporters [27], suggesting that putrescine increased amino acid accumulation in muscle might be via promoting amino acid transport. Collectively, we found that dietary 0.03% and 0.05% putrescine supplementation increased the quality of chicken meat.

A nutrition strategy provides an effective way for consumers to obtain healthy chicken products. The current study revealed that dietary supplementation with putrescine could modulate the fatty acid composition and decrease the content of SFA in the breast meat of Wenchang chickens. Cardiovascular disease is considered closely associated with an excess SFAs intake [28]. In the present study, the contents of SFAs in the breast muscle were significantly decreased, which indicated that the meat from the broilers treated with putrescine was more beneficial to human health. It was demonstrated that a higher n-3 PUFAs level or lower n-6/n-3 ratio reduced the risk of cardiovascular diseases and cancer incidence [29]. In the present study, dietary 0.05% putrescine supplementation decreased the n-6 PUFAs content and n-6/n-3 PUFA ratio in muscle by 41.08% and 30.39%, respectively. However, studies about the effect of polyamine on fatty acid metabolism mainly focus on spermidine. For instant, Ma et al. [30] stated that spermidine supplementation causes a significant loss of weight and improves insulin resistance in diet-induced obese mice. Zhou et al. [31] reported that spermidine supplementation increases fatty acid oxidation and reduces hepatic lipid species in the liver of mice with non-alcoholic fatty liver disease. Collectively, our research provided evidence that putrescine can regulate fatty acid metabolism in the muscle of chickens.

To our knowledge, this is the first study to investigate the effects of putrescine on the IMP and elemental composition of broiler muscle. It has been demonstrated that the presence of putrescine in broiler chicken diets, particularly at the levels of 0.01% and 0.03%, increased the Na content in the breast muscle. It is widely accepted that Na is an essential nutrient to maintain proper blood volume and pressure [32]. The results implied that the inclusion of 0.05% putrescine in diets did not impede mineral deposition in muscle.

## 5. Conclusions

In conclusion, the present study showed that dietary supplementation with 0.05% putrescine improved the growth performance, feed conversion ratio, amino acid content and fatty acid profile of Wenchang chickens, which suggested that dietary supplementation with 0.05% putrescine is an effective nutrition strategy to improve the growth performance and meat quality of Wenchang chickens.

## Figures and Tables

**Figure 1 animals-13-01564-f001:**
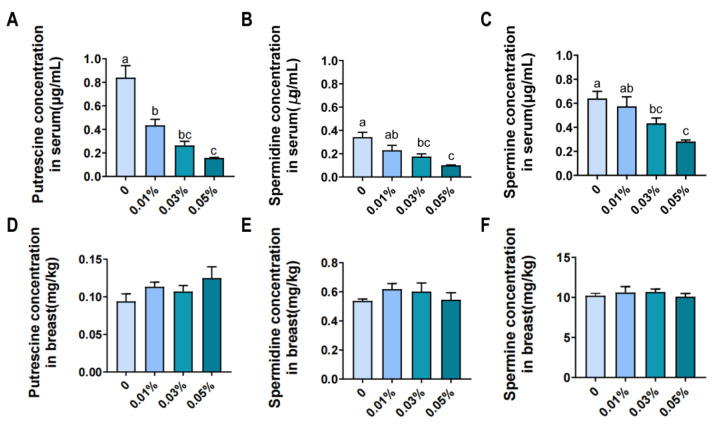
The contents of putrescine, spermidine, and spermine in serum (**A**–**C**) and breast muscle (**D**–**F**) of Wenchang chickens. The values are the means ± SEM, *n* = 8. The different superscript letters (a–c) above the bars show significant difference at *p* < 0.05.

**Figure 2 animals-13-01564-f002:**
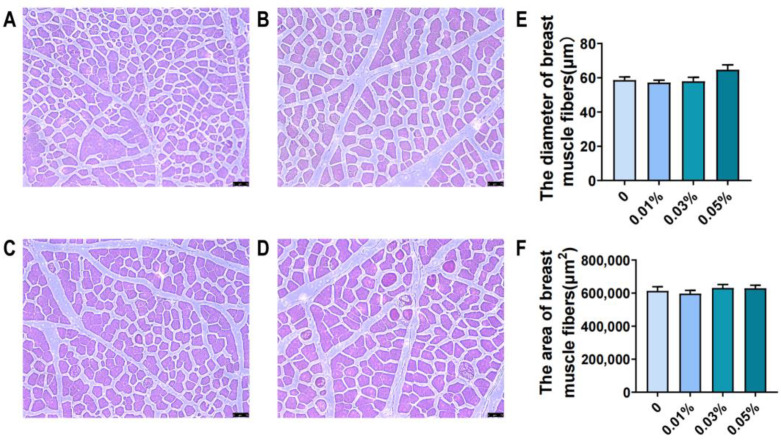
The morphology of breast muscle. (**A**–**D**): Control, 0.01% putrescine group, 0.03% putrescine group, and 0.05% putrescine group, respectively. The diameter (**E**) and area (**F**) of muscle fibers of Wenchang chickens. The values are the means ± SEM, *n* = 8, bar = 75 μm.

**Table 1 animals-13-01564-t001:** Composition and nutrient level of diets (dry matter basis, %).

Ingredients	Control
Corn	66.91
Soybean meal	22.00
Lysine	0.09
Soybean oil	3.50
Methionine	0.10
Bran	4.10
Limestone	1.30
Dicalcium phosphate	0.70
Nacl	0.30
Premix ^1^	1.00
Analyzed nutrient content	
ME, kcal/kg	3049.58
Crude protein	16.00
Crude Fat	6.42
Lysine	0.96
Methionine	0.26
Calcium	0.75
Phosphorous	0.47
Putrescine, mg/kg	1.02

^1^ Provided per kilogram of diet: Copper 10 mg, Iron 80 mg, Manganese 60 mg, Zinc 70 mg, Iodine 2 mg, Selenium 0.40 mg, vitamin A 10,000 IU, vitamin D_3_ 2000 IU, vitamin E 10 mg, vitamin B_1_ 2 mg, vitamin B_2_ 3 mg, vitamin B_6_ 3.50 mg, nicotinic acid 15 mg, folic acid 0.50 mg, pantothenic acid 10 mg, biotin 0.15 mg, cyanocobalamin 10 μg.

**Table 2 animals-13-01564-t002:** Effects of dietary supplementation with putrescine on growth performance of Wenchang Chickens.

Items	Putrescine Concentration				SEM	*p*-Value
	Con	0.01%	0.03%	0.05%		
IW (g)	1277.50	1277.50	1279.58	1278.33	0.72	0.58
FW (g)	1917.98 ^b^	1934.08 ^b^	1948.01 ^ab^	1990.33 ^a^	8.51	0.01
ADFI (g)	96.24	92.49	96.51	97.71	0.86	0.08
ADG (g)	16.01 ^b^	16.41 ^b^	16.71 ^ab^	17.80 ^a^	0.21	0.01
F:G (g/g)	6.02 ^a^	5.64 ^b^	5.80 ^ab^	5.49 ^b^	0.06	0.02

^a, b^ Mean within rows with different letters differ significantly (*p* < 0.05), *n* = 8. IW: initial weight, FW: final wight, ADFI: average daily feed intake, ADG: average daily weight gain, F:G, the ratio of feed intake to body weight gain.

**Table 3 animals-13-01564-t003:** Effects of dietary supplementation with putrescine on carcass traits of Wenchang Chickens (%).

Items	Putrescine Concentration				SEM	*p*-Value
	Con	0.01%	0.03%	0.05%		
Carcass yield	91.44	91.90	90.74	91.60	0.24	0.62
Semi-eviscerated	79.12	80.67	78.73	79.60	0.51	0.71
Eviscerated	64.28	65.77	63.83	64.98	0.46	0.49
Abdominal fat	7.95	8.09	8.28	8.25	0.30	0.98
Breast muscle	15.16	15.57	15.49	14.19	0.23	0.11
Thigh muscle	18.25	17.73	18.81	18.23	0.26	0.57

**Table 4 animals-13-01564-t004:** Effects of dietary supplementation with putrescine on meat quality of Wenchang Chickens.

Items	Putrescine Concentration				SEM	*p*-Value
	Con	0.01%	0.03%	0.05%		
pH_45min_	6.36	6.32	6.28	6.32	0.05	0.98
pH_24h_	5.74	5.73	5.75	5.78	0.02	0.92
Lightness (L*)	49.56	50.94	51.18	50.59	0.91	0.94
Yellowness (b*)	10.59	13.06	13.02	12.73	0.57	0.38
Drip loss (%)	4.64	5.64	5.64	5.69	0.48	0.83
Shear force (N/cm^2^)	44.43	50.35	44.28	41.42	1.68	0.37

**Table 5 animals-13-01564-t005:** The contents of amino acids in breast meat of Wenchang chickens (dry matter basis, mg/g).

Items	Putrescine Concentration				SEM	*p*-Value
	Con	0.01%	0.03%	0.05%		
EAA						1
Val	43.46	43.25	49.57	47.13	1.16	0.15
Thr	30.61 ^b^	32.86 ^ab^	35.67 ^a^	33.63 ^ab^	0.62	0.02
Met	19.49 ^b^	21.39 ^b^	22.75 ^a^	21.30 ^a^	0.41	0.01
lle	32.03	33.63	38.83	36.53	0.97	0.06
Leu	56.03	59.50	67.01	63.81	1.69	0.10
Phe	23.48 ^c^	25.60 ^bc^	28.05 ^a^	26.30 ^ab^	0.66	<0.01
Lys	80.05 ^c^	88.93 ^bc^	94.38 ^ab^	96.51 ^a^	2.31	<0.01
Trp	0.30	0.29	0.29	0.28	0.01	0.88
NEAA						
Gly	26.67	28.87	28.72	25.73	0.73	0.22
Ser	20.95	22.40	21.24	20.20	0.64	0.82
Ala	46.53	49.89	54.82	48.54	1.56	0.07
Asp	59.50 ^c^	64.95 ^bc^	68.71 ^a^	66.55 ^ab^	1.60	0.01
Pro	26.80 ^b^	29.93 ^b^	32.63 ^a^	29.36 ^b^	0.85	0.01
Tyr	16.40 ^b^	18.94 ^ab^	20.47 ^a^	20.68 ^a^	0.64	<0.01
Arg	33.69	35.47	31.54	34.16	1.41	0.34
Cys	3.25	3.50	3.30	2.82	0.13	0.52
Glu	77.97	86.22	85.56	77.28	2.35	0.23
His	30.63	34.96	40.01	35.46	1.44	0.15
∑EAAs	285.44 ^b^	304.37 ^b^	333.71 ^a^	321.29 ^a^	8.04	<0.05
∑NEAAs	342.39	375.14	387.00	360.78	9.09	0.06
∑TAAs	627.82 ^c^	679.50 ^bc^	720.70 ^a^	682.07 ^ab^	16.77	0.01

EAAs: essential amino acids; NEAAs: nonessential amino acids; TAAs: total amino acids. ^a–c^ Mean within rows with different letters differ significantly *(p* < 0.05).

**Table 6 animals-13-01564-t006:** The contents of fatty acids in breast muscle of Wenchang chickens (dry matter basis, μg/g).

Items	Putrescine Concentration				SEM	*p*-Value
	Con	0.01%	0.03%	0.05%		
C6:0	0.47	0.39	0.41	0.40	0.02	0.21
C8:0	1.20	0.83	0.67	0.71	0.08	0.13
C10:0	3.79	2.89	2.24	2.46	0.22	0.06
C11:0	0.26 ^a^	0.17 ^b^	0.13 ^b^	0.13 ^b^	0.01	<0.01
C12:0	15.74 ^a^	11.62 ^ab^	9.99 ^b^	10.00 ^b^	0.81	<0.05
C13:0	0.80 ^a^	0.55 ^ab^	0.48 ^b^	0.45 ^b^	0.05	0.03
C14:0	330.51 ^a^	240.64 ^ab^	204.84 ^b^	205.97 ^b^	18.63	<0.05
C15:0	42.04 ^a^	31.68 ^ab^	27.19 ^b^	25.50 ^b^	2.21	0.04
C16:0	22,276.10	16,775.93	13,564.91	13,876.99	1299.36	0.10
C17:0	91.61	67.10	55.25	53.39	5.27	0.07
C18:0	6606.14	5018.99	4357.17	4559.51	322.32	0.08
C20:0	53.39 ^a^	38.93 ^ab^	32.54 ^b^	30.43 ^b^	3.14	<0.05
C21:0	23.82	22.12	19.50	21.33	0.56	0.06
C22:0	11.75 ^a^	9.58 ^ab^	8.05 ^b^	7.37 ^b^	0.60	<0.05
C23:0	10.65 ^a^	9.99 ^ab^	9.93 ^b^	9.62 ^b^	0.12	0.01
C24:0	5.62 ^a^	3.39 ^ab^	2.66 ^bc^	2.37 ^c^	0.33	<0.01
C14:1	92.81	88.98	69.41	62.11	7.36	0.40
C16:1	3029.32	2739.30	1996.98	1850.24	213.05	0.15
C17:1	58.62	47.24	37.74	36.01	3.49	0.09
C18:1n9c	37,349.91	29,527.51	22,638.06	23,216.27	2355.38	0.14
C20:1	238.58	194.33	144.19	156.82	14.56	0.16
C22:1n9	14.41 ^a^	12.10 ^ab^	9.58 ^b^	8.93 ^b^	0.78	0.04
C20:3n3	24.99 ^a^	24.48 ^a^	20.45 ^b^	21.52 ^b^	0.56	<0.01
C20:5n3	20.68	24.60	18.92	19.34	1.45	0.52
C18:3n3	703.76	667.10	495.24	467.31	46.65	0.15
C22:6n3	308.88	342.19	329.28	419.07	18.05	0.22
C20:3n6	266.87 ^a^	251.97 ^a^	211.61 ^b^	246.82 ^ab^	7.21	0.04
C20:4n6	2785.92	2559.33	2565.71	2789.90	59.60	0.31
C20:2n6	221.08	199.63	166.20	178.60	9.12	0.11
C18:2n6c	26,014.85 ^a^	17,999.73 ^ab^	14,821.44 ^b^	14,034.60 ^b^	1611.01	<0.05
C18:3n6	143.64	94.45	97.56	91.23	9.21	0.14
∑SFAs	29,464.88	22,234.80	18,295.97	18,806.64	1643.20	0.11
∑MUFAs	40,783.65	32,609.46	24,895.97	25,330.37	2572.26	0.14
n-3PUFAs	1058.31	1058.37	863.89	927.25	51.87	0.47
n-6PUFAs	29,432.35 ^a^	21,105.12 ^ab^	17,862.53 ^b^	17,341.14 ^b^	1645.07	<0.05
n-6/n-3PUFAs	27.05^a^	20.00 ^b^	20.43 ^b^	18.83 ^b^	0.71	<0.01

^a–c^ Mean within rows with different letters differ significantly (*p* < 0.05). SFAs: saturated fatty acids; MUFAs: monounsaturated fatty acids; PUFAs: polyunsaturated fatty acids.

**Table 7 animals-13-01564-t007:** The IMP and elemental composition in breast meat of Wenchang chickens (dry matter basis, mg/100 g).

Items	Putrescine Concentration				SEM	*p*-Value
	Con	0.01%	0.03%	0.05%		
IMP	10.12	12.21	8.98	8.46	0.99	0.57
K	1306.53	1404.53	1550.31	1308.13	0.44	0.17
Mg	116.34	126.56	130.7	112.04	0.03	0.18
Na	127.55 ^b^	162.09 ^a^	171.69 ^a^	130.63 ^b^	0.06	0.01
Zn	1.42	1.57	1.58	1.42	0.44	0.41

^a,b^ Mean within rows with different letters differ significantly (*p* < 0.05).

## Data Availability

The original contributions generated for this study are included in the article. Further inquiries can be directed to the corresponding author.

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
