# Peer review of "Dietary Supplementation with Putrescine Improves Growth Performance and Meat Quality of Wenchang Chickens"

_animals, 2023, doi:10.3390/ani13091564_

Round 1

Reviewer 1 Report

1.      The simple summary is suggested to rewrite to briefly describe aims, major findings, and conclusions instead of looking like introduction.

2.      Grammars and writing should be checked thoroughly and corrected. Here are some examples in the Simple Summary and Abstract.

Ex. line 15-17, a useful feed additive for animal production should not be limited, not only functioning in growth efficiency, but also improving carcass quality. birth …

EX. Line 17-19, Putrescine is receiving attentions for its antioxidant capacity, but but little emphasis has been placed on the effect of putrescine to relieve oxidative stress on meat quality.

EX. Line 19-21, This study demonstrated that putrescine supplementation is an effective nutrition….and meat quality by increasing the content of essential amino acids and reducing saturated fatty acid concentrations.

EX. Line 22, that supplemental putrescine is effective to prevent meat quality decline in chicken meat.

EX. Line 30-31, The concentrations of serum putrescine, spermidine, and spermine were lower (P < 0.05) in birds with putrescine supplementations.

EX. Line 32, total essential amino acids, and total amino acids…

EX. Line 36, total n-6 monounsaturated fatty acids

EX. Line 38, n-6/n-3 polyunsaturated fatty acids…. Overall, dietary supplementation

3.      Why bran was use to formulate the feed? Decreasing the CP%?

4.      The authors need to justify why CP only account for 16% in feed composition?  

5.      Why 80-day old birds instead of day-old were used for the study.

6.      Table 2 title, … supplemental putrescine….

7.      Figure 1 title, Effects of supplemental putrescine on serum (A-C) and breast muscle (D-F) putrescine, spermidine, and spermine content of Wenchang chickens.

8.      Putrescine is volatile with a pungent unpleasant odor. What is the excipient or how the authors particulate putrescine into feed?  

Author Response

Point 1: The simple summary is suggested to rewrite to briefly describe aims, major findings, and conclusions instead of looking like introduction.

Reply:  We appreciate the reviewer’s comment. The simple summary was rewritten as you suggested. Please see Line15-21.

Point 2: Grammars and writing should be checked thoroughly and corrected. Here are some examples in the Simple Summary and Abstract.

Ex. line 15-17, a useful feed additive for animal production should not be limited, not only functioning in growth efficiency, but also improving carcass quality. birth …

  1. Line 17-19, Putrescine is receiving attentions for its antioxidant capacity, but but little emphasis has been placed on the effect of putrescine to relieve oxidative stress on meat quality.
  2. Line 19-21, This study demonstrated that putrescine supplementation is an effective nutrition….and meat quality by increasing the content of essential amino acids and reducing saturated fatty acid concentrations.
  3. Line 22, that supplemental putrescine is effective to prevent meat quality decline in chicken meat.
  4. Line 30-31, The concentrations of serum putrescine, spermidine, and spermine were lower (P < 0.05) in birds with putrescine supplementations.
  5. Line 32, total essential amino acids, and total amino acids…
  6. Line 36, total n-6 monounsaturated fatty acids
  7. Line 38, n-6/n-3 polyunsaturated fatty acids…. Overall, dietary supplementation

Reply:  Thank you for the valuable comments. Grammars and writing have been checked thoroughly and corrected as you suggested, please see the revised manuscript.

Point 3: Why bran was use to formulate the feed? Decreasing the CP%?

Reply:  Thank you for your comments. Bran is one of the most used protein ingredients in feed for broilers. Bran used in the feed could reduce the content of soybean meal in feed.

Point 4: The authors need to justify why CP only account for 16% in feed composition? 

Reply:  Thank you for the valuable comments. The recommended crude protein level in feed for yellow-feathered broilers of 8 weeks of age is 16% according to the Nutrient Requirements of Yellow-feathered Broiler (Ministry of Agriculture, 2004) (NY/T 33-2004). Based on this principle, the crude protein of the feed was designed as 16%.

Point 5: Why 80-day old birds instead of day-old were used for the study.

Reply:  Thank you for your comments. 80-120 days of age is the rapid muscle development period for Wenchang chickens. It is more significant to study the effects of putrescine on the growth performance and meat quality of Wenchang chickens during this period.

Point 6: Table 2 title, … supplemental putrescine….

Reply:  Table 2 title has been corrected in manuscript. (Line 159-160)

Point 7: Figure 1 title, Effects of supplemental putrescine on serum (A-C) and breast muscle (D-F) putrescine, spermidine, and spermine content of Wenchang chickens.

Reply:   Figure 1 title has been corrected in manuscript. (Line 172-173)

Point 8: Putrescine is volatile with a pungent unpleasant odor. What is the excipient or how the authors particulate putrescine into feed?

Reply:  Thank you for your comments. Considering that putrescine is volatile with a pungent odor, we added putrescine hydrochloride to the feed which is more stable and less smelly.

Reviewer 2 Report

Subject - Is the subject of the work a question or a statement? It should be redrafted

Introduction

Line 47 - why is the meat quality reduced?

Line 49 - I don't understand this sentence, what did the Author mean?

Line 53-60 - should be reworded, give the advantages and disadvantages of using putrescine in animal and human models, taking into account primarily the use in poultry

There is no research hypothesis

Materials and Methods

1. There are no exact data on chickens, were they vaccinated against infectious diseases? Where did they come from?

2. Table 1 - Were the chickens fed one type of feed from birth to kill? Where's the starter, grower, finisher? They all got putrescine. The table is illegible and requires correction. In addition, it is not possible in intensive farming to use one type of feed throughout the rearing period.

3. How many birds were blood drawn from? How many birds were tissues taken from? Complete all sections with this information.

4. Figure - keep the spacing in the axis captions

Discussion

1. Please do not describe the results in the discussion - please edit the text

2. It should be clarified at the beginning that putrescine belongs to polyamines, otherwise the text is unclear

3. Part of the discussion is just a repetition of the Results chapter - this should be removed

4.  Please draw and describe the conclusions from the obtained results

conclusion

1. Delete the last sentence

References

1. Keep the recommended layout for the magazine, e.g. 18 is incorrect

Author Response

Subject

Point 1: Is the subject of the work a question or a statement? It should be redrafted

Reply:  The simple summary was rewritten according to your comment. (Line15-21)

Introduction

Point 2: Line 47 - why is the meat quality reduced?

Reply:  Stress could induce the production of harmful substances, such as MDA and ROS, thus resulting in the decline of meat quality.

Zaboli G, Huang X, Feng X, Ahn DU. How can heat stress affect chicken meat quality? - a review. Poult Sci. 2019 Mar 1;98(3):1551-1556;

 Xing T, Gao F, Tume RK, Zhou G, Xu X. Stress Effects on Meat Quality: A Mechanistic Perspective. Compr Rev Food Sci Food Saf. 2019 Mar;18(2):380-401.

Point 3: Line 49 - I don't understand this sentence, what did the Author mean?

Reply:  This sentence was deleted according to your comment.

Point 4: Line 53-60 - should be reworded, give the advantages and disadvantages of using putrescine in animal and human models, taking into account primarily the use in poultry

Reply:  Thank you for the valuable comments. The discussion was reworded as you suggested. Please see line 52-60.

Point 5: There is no research hypothesis

Reply:  Research hypotheses have been supplemented in the manuscript. Please see line 61-62.

Materials and Methods

Point 6: There are no exact data on chickens, were they vaccinated against infectious diseases? Where did they come from?

Reply:  The chickens used in our experiment were routinely immunized. The information was added to the revised manuscript. Please see line 74-75.

Point 7: Table 1 - Were the chickens fed one type of feed from birth to kill? Where's the starter, grower, finisher? They all got putrescine. The table is illegible and requires correction. In addition, it is not possible in intensive farming to use one type of feed throughout the rearing period.

Reply:  Eighty-day-old female broilers (Wenchang chickens) were fed with the basic diet listed in Table 1. Before 80 days of age, broilers were fed a basal corn-soybean meal diet that met the nutritional requirements of the growing stage, which is different from the basic diet used in this study.

Point 8: How many birds were blood drawn from? How many birds were tissues taken from? Complete all sections with this information.

Reply: At the end of experiment, 8 chickens close to final BW were selected for blood and muscle collection in each group. The information was added to the revised manuscript. Please see line 87-88.

Point 9:  Figure - keep the spacing in the axis captions

Reply:  The spacing in the axis captions of Figure1. (A-C) have been corrected in manuscript, but the concentrations of polyamines in the breast muscle varied greatly, and the spacing in the axis captions of Figure1.(D-F) unchanged.

Discussion

Point 10:  Please do not describe the results in the discussion - please edit the text

Reply: Thank you for your valuable comments. The discussion was revised as you suggested, the repetition of the results chapter have been removed in manuscript.

Point 11:  It should be clarified at the beginning that putrescine belongs to polyamines, otherwise the text is unclear

Reply:  Thank you for your valuable comments. The sentence of “Putrescine is a main polyamine found in animals, which plays an important role in cell development and protein synthesis.” was added to the manuscript. Please see line 52-53.

Point 12:  Part of the discussion is just a repetition of the Results chapter - this should be removed

Reply:   Thank you for your valuable comments. The discussion was revised as you suggested, the repetition of the results chapter have been removed in manuscript. 

Point 13:  Please draw and describe the conclusions from the obtained results

Reply:  We appreciate the reviewer’s comment. We have revised the conclusions, please see line 298-302.

conclusion

Point 14:  Delete the last sentence

Reply:  The last sentence has been deleted as you suggested.

Point 15:  Keep the recommended layout for the magazine, e.g. 18 is incorrect

Reply:  Thank you for your valuable comments. The format of the reference was revised as the magazine recommended, please see the revised manuscript.

Reviewer 3 Report

Line 70: Basal rather than basial

Line 70-71: The authors need to specify what type of broiler breed nutrient specification they have used.

Table 1 (line 78): Can the authors double check if a basal diet supplemented with 22% soybean meal provided 16% crude protein? The authors may also need to provide what type of bran used.

The authors may also need to reduce overuse of person pronoun in the manuscript (line 222; 246; 261; 267; 284; and 288).

Author Response

Point 1: Line 70: Basal rather than basial

Reply: It has been corrected in the manuscript as you suggested. Please see line 71.

Point 2: Line 70-71: The authors need to specify what type of broiler breed nutrient specification they have used.

Reply: The basal diet (Table 1) was designed in accordance with the Nutrient Requirements of Yellow-feathered Broiler (Ministry of Agriculture, 2004). Details was added to the manuscript. Please see line 71-73.

Point 3: Table 1 (line 78): Can the authors double check if a basal diet supplemented with 22% soybean meal provided 16% crude protein? The authors may also need to provide what type of bran used.

Reply:  Thank you for the valuable comments. The crude protein contents of corn, soybean meal, and bran is 8.5%, 44%, and 15.5%, respectively. The crude protein of the feed was calculated as follows:

CP (%) = 66.91%×8.5% + 22%×44% + 4.1%×15.5%=16%

Point 4:  The authors may also need to reduce overuse of person pronoun in the manuscript (line 222; 246; 261; 267; 284; and 288).

Reply:  Many thanks for the reviewer’s comment. The use of personal pronouns has been reduced in the manuscript according to your comment. Please see the revised manuscript.

Round 2

Reviewer 2 Report

I have no comments